# Is It Possible to Access the Uterus of Sheep by Endoscopy: Studies of Vaginoscopy and Hysteroscopy with Transcervical Uterine Access in Sheep

**DOI:** 10.3390/life15060846

**Published:** 2025-05-23

**Authors:** Augusto Ryonosuke Taira, Thiago da Silva Cardoso, Renata Levy Amanajas, Renata Sitta Gomes Mariano Landers, Priscila Del Águila da Silva, Victor Jóse Correia Santos, Naiara Nantes Rodrigues, Dayane Priscila Vrisman, Felipe Faria Pereira da Câmara Barros, Francisco Décio de Oliveira Monteiro, Rodrigo dos Santos Albuquerque, Felipe Masiero Salvarani, Wilter Ricardo Russiano Vicente, Pedro Paulo Maia Teixeira

**Affiliations:** 1Faculdade de Ciências Agrárias e Veterinária, Universidade Estadual Paulista, Jaboticabal 14884-900, SP, Brazil; augusto.vete@gmail.com (A.R.T.); rlanders@tamu.edu (R.S.G.M.L.); priscila.aguila@gmail.com (P.D.Á.d.S.); santosvjc@gmail.com (V.J.C.S.); naiara_nantes@hotmail.com (N.N.R.); dayavrisman@hotmail.com (D.P.V.); wilterrussiano@gmail.com (W.R.R.V.); 2Instituto de Medicina Veterinária, Universidade Federal do Pará, Castanhal 68740-970, PA, Brazil; thiagodacardoso09@gmail.com (T.d.S.C.); renata.amanajas@gmail.com (R.L.A.); deciomonteiro@ifto.edu.br (F.D.d.O.M.); rdsa20@gmail.com (R.d.S.A.); 3Instituto de Medicina Veterinária, Universidade Federal Rural do Rio de Janeiro, Seropédica 23890-000, RJ, Brazil; felipebarros@ufrrj.br

**Keywords:** cervix, sheep, cervical traction, vaginoscopy, endoscopy

## Abstract

Given the endoscopic possibilities and the need to improve AI in sheep, the aim of this study was to develop a transcervical endoscopic technique for accessing the uterus in sheep. The study was conducted on 35 ewes divided according to the uterus accessing technique applied for artificial insemination (AI). In a pilot study, two techniques were tested using a rigid endoscope coupled to a protective sheath totaling 3 mm, in a group of ewes not subjected to fixed-time artificial insemination (FTAI) protocol and those subjected to the protocol (EPG, *n* = 5 and EPGp, *n* = 10). After the pilot study, two additional techniques were tested in synchronized ewes for FTAI: a control group with cervical traction (CG, *n* = 10) and an AI group using vaginoscopy with a multiport for the passage of a rigid endoscope (EVG, *n* = 10). The EPG and EPGp showed 100% (5/5) and 10% (1/10) cervical passage rates, respectively. The EPGp had 90% (9/10) superficial cervical inseminations, 10% (1/10) intrauterine inseminations, and a 10% (1/10) pregnancy rate. In CG and EVG, 3.5 ± 3.3 and 1.6 ± 1.2 cervical rings were passed, respectively. Additionally, semen deposition resulted in 20% (2/10) intrauterine inseminations and 80% (8/10) deep cervical inseminations for CG, while EVG had 20% (2/10) intrauterine inseminations and 80% (8/10) superficial cervical inseminations. The pregnancy rate was 20% (2/10) for both CG and EVG. The EPG technique proved efficient for hysteroscopy; however, EPGp was not efficient for AI due to the presence of typical estrus mucus. Nevertheless, it laid the foundation for the development of EVG, which showed promise in gynecological evaluations, enabling intrauterine AI and a complete gynecological assessment.

## 1. Introduction

Among the reproductive biotechnologies used in sheep farming, artificial insemination (AI) stands out for effectively contributing to genetic improvement. However, transcervical access to the uterus in ewes is a limiting factor for AI, as unlike cattle, rectal manipulation is not possible, and ewes have a narrow cervical ostium with rigid and tortuous structures, making the passage of the insemination pipette difficult and limiting the technique’s application [1,2]. In sheep, video-assisted surgery is efficiently employed in repeated follicular aspirations and artificial insemination via laparoscopy and as a diagnostic method for reproductive abnormalities and diseases through vaginoscopy. However, it still needs to be studied for hysteroscopy [3,4].

The development of minimally invasive techniques in reproductive medicine has significantly transformed both human and veterinary gynecology, enabling procedures with greater accuracy, reduced trauma, and improved outcomes [5]. In human medicine, hysteroscopic approaches to endometrial access and intervention such as directed biopsies and ablations have demonstrated superiority over blind procedures by increasing diagnostic precision and minimizing complications [6]. These principles are equally applicable to veterinary medicine, where adapting and validating such techniques in small ruminants can improve fertility rates while ensuring animal welfare [5,6].

Furthermore, recent guidelines and systematic reviews emphasize the importance of tailoring endoscopic techniques to anatomical variability, using appropriate instrumentation and visualization systems to optimize diagnostic and therapeutic success [7]. Transposing these recommendations to veterinary settings, particularly in species like sheep with complex cervical anatomy, underscores the relevance of studies aimed at refining transcervical procedures [8]. Thus, exploring endoscopic-assisted insemination aligns with a broader scientific effort to enhance reproductive efficiency through precision, safety, and innovation [7,8].

To optimize reproduction in ruminants, new endoscopic techniques for vaginoscopy have been developed, allowing better vaginal evaluation and even transcervical passage without rectal palpation [9]. Given these endoscopic possibilities and the need to improve AI in sheep, the present study aimed to develop endoscopic techniques for videovaginoscopy and hysteroscopy, as well as transcervical artificial insemination in sheep.

## 2. Materials and Methods

### 2.1. Experimental Animals

The animal study protocol number 6261300323 (ID 002208) was approved by the National Council for Control of Animal Experimentation and was approved by the Ethics Committee on Animal Use of the Federal University of Para. Thirty-five multiparous Santa Inês and crossbred Santa Inês ewes, aged 3 to 5 years, with a body condition score of approximately 3 on a 1–5 scale [10], and an average body weight of 45 ± 4.6 kg were used. The animals were deemed healthy following clinical and ultrasonographic examination of their reproductive organs. The ewes were housed in pens with ad libitum access to water and mineral salt, and were fed a balanced diet (corn silage/commercial concentrate—14% CP and 65% TDN) according to their experimental group.

### 2.2. Pilot Study of Access to the Uterus

Initially, a pilot study was conducted with five ewes not subjected to a fixed-time artificial insemination (FTAI) protocol, in anestrus, which underwent endoscopic access to the uterus (Endoscopic Pilot Group without protocol—EPG, *n* = 5). The procedure was performed using a rigid endoscope (Karl Storz, Tuttlingen, Germany) of 2.7 mm diameter and 17.5 cm length, with a 10° viewing angle, coupled to a 3 mm working sheath, connected to a video surgery system. The ewes were physically restrained with their hind limbs elevated. A vaginal speculum was used to visualize the cervix, followed by the instillation of 2 mL of 2% lidocaine hydrochloride into the vaginal fornix, with a one-minute wait before the procedure. The cervix was anchored using 25 cm Allis forceps (Karl Storz, Tuttlingen, Germany), and the endoscope was introduced through the ostium with the aim of passing through the cervix and entering the uterus. The endoscope was then withdrawn, leaving its sheath in place to allow the passage of the insemination pipette; however, insemination was not performed.

### 2.3. Second Pilot Study of Uterine Access with Fixed-Time Artificial Insemination (FTAI) Protocol

Subsequently, a second pilot study was conducted with ten additional multiparous ewes subjected to the same AI technique as EPG but now under a standard short FTAI protocol (Endoscopic Pilot Group with protocol—EPGp *n* = 10). The protocol consisted of six days of progesterone (0.33 g of progesterone—CIDR^®^, Pfizer, São Paulo, Brazil), associated with prostaglandin (37.5 μg of cloprostenol sodium—Sincrocio^®^, Ourofino, São Paulo, Brazil) and equine chorionic gonadotropin (300 IU of eCG—Novormon^®^, Zoetis, São Paulo, Brazil). Inseminations were performed 48 to 55 h after progesterone device removal using thawed semen from a ram. If all cervical rings were passed or five minutes of manipulation elapsed, the endoscope was withdrawn, and semen was deposited using the protective sheath at the site of maximum progression. Pregnancy confirmation was conducted 21 days post-AI via transrectal ultrasonography using a MyLabTM 30VET device (Esaote S.p.A., Genoa, Liguria, Italy) with a 7.5 MHz linear transducer.

At the end of this phase, the frequency distribution of cervical passage, pregnancy rates, and relevant observations were recorded. Data were analyzed using the chi-square test, with absolute and percentage data processed using the Bioestat 5.3 statistical package.

### 2.4. Study After Adaptations of the Previous Phase

After evaluating the results of the pilot phase, two groups of ewes were used. These ewes underwent the same FTAI protocol as EPGp. They were divided into two groups: conventional FTAI as the control group with cervical traction (CG, *n* = 10) and endoscopic vaginoscopy with vaginal insufflation and no cervical fixation (Endoscopic Vaginoscopy Group—EVG, *n* = 10).

For CG, the hind limbs of the animal were elevated onto an 80 cm high stand, and the perineal region was cleaned beforehand. A 15 cm vaginal speculum (Karl Storz, Tuttlingen, Germany) was then used to open and observe the cervix. Additionally, 2 mL of 2% lidocaine hydrochloride was instilled into the vaginal fornix, with a one-minute wait before the procedure. With the aid of an external light source, the cervix was visualized, classified, and grasped using a 25 cm Allis forceps. The cervix was then placed in traction to the vaginal vestibule, and an ovine applicator (Karl Storz, Tuttlingen, Germany) (13 cm long and 2 mm in diameter) was used for cervical passage. When the uterine interior was reached or after three minutes of cervical manipulation, thawed semen was deposited using the semen applicator (Karl Storz, Tuttlingen, Germany) at the deepest accessible location.

For GEV, FTAI was performed with the animal in a quadrupedal position, following the same antisepsis procedure as in the other groups, but without local anesthesia. A multi-access portal (Multiport, Sitracc Less, Edlo) was inserted into the vaginal vestibule with the aid of a non-spermicidal lubricant. After positioning the multiport, it was connected to the video surgery insufflation system, promoting vaginal cavity insufflation with carbon dioxide (CO_2_) at a flow rate of 6 L/min at a pressure of 10 mmHg. A rigid endoscope (5 mm in diameter, 17.5 cm in length, and 0° viewing angle) connected to the video surgery system was used to evaluate the vaginal cavity and cervical ostium. Subsequently, a commercial 31 cm caprine semen applicator was inserted into one of the working channels of the multiport trocar for cervical passage. Semen deposition was performed as soon as all cervical rings were passed or after three minutes of cervical manipulation (Figure 1).

For this stage, cervical ostia were classified according to Kershaw et al. [11] as follows: duckbill, flap, smooth, papilla, or rosette. The number of rings passed was measured according to the progression of the applicator or endoscope, and semen deposition location was classified based on the number of rings passed, categorized according to Taqueda et al. [12] as superficial cervical (up to the 3rd ring), deep cervical (beyond the 3rd ring), and intrauterine, thus determining the cervical passage rate.

Serum samples were also used to evaluate inflammatory response at seven time points: 5 min before AI, and at 20 min, 24, 48, 72, 96, and 120 h post-AI. Samples were obtained via jugular venipuncture using vacuum tubes containing EDTA (Vacutainer, BD, São Paulo, Brazil), immediately centrifuged at 1500× *g* for 10 min. Plasma was collected and stored in microtubes at −80 °C until analysis. Plasma fibrinogen was analyzed using the method described by Millar et al. [13] through heat precipitation at 56 °C and manual refractometer reading, with values expressed in mg/dL. Pregnancy diagnosis was performed using the same methodology as in the pilot.

Collected data were subjected to normality testing (Shapiro test) [14]. The number of rings passed, semen deposition site, and plasma fibrinogen levels were compared between AI techniques using the Kruskal–Wallis test [15] and Dunn’s post hoc test [16]. Cervical manipulation time was compared using ANOVA and Tukey’s post hoc test [17]. Pregnancy rate was compared using the chi-square test [18], and fibrinogen variation was calculated as a percentage change from baseline (considered as 0). Analyses were performed using Bioestat 5.3, with significance set at *p* < 0.05.

## 3. Results

During the initial study phase, cervical passage was successful in 100% of the EPG (5/5 sheep) and only 10% of the EPGp group (1/10 sheep). In the EPGp group, 90% of inseminations (9/10) were superficial cervical, while only 10% (1/10) reached the uterus (intrauterine insemination). The only female that became pregnant in this group was the one that received intrauterine insemination, resulting in a pregnancy rate of 10%. Conversely, all sheep in the EPG group had visible cervical trajectories, allowing successful instrument passage. In the EPGp group, this passage was hindered by whitish secretions within the cervical lumen characteristic of estrus, preventing uterine access in 90% of synchronized females. This finding led to the development of the EVG technique, which was compared to the conventional CG technique (Table 1). The time required for insemination was significantly greater in the EPGp (249 ± 101 s) compared to the CG (158 ± 50) and EVG (129 ± 29) groups, which did not differ from each other (*p* = 0.0016), as shown in Table 1.

The number of cervical rings traversed during artificial insemination was significantly greater in the CG (3.5 ± 3.3) compared to the EVG (1.6 ± 1.2; *p* = 0.0065). Regarding semen deposition location, 20% of inseminations were intrauterine and 80% were deep cervical in the CG group. In the EVG, the same percentages were observed for intrauterine (20%) and superficial cervical (80%) inseminations (Table 2).

Plasma fibrinogen analysis revealed time-dependent dynamics. While all values remained within the species’ physiological range (100–500 mg/dL), EVG exhibited a transient fibrinogen increase at 24 h post-insemination (*p* = 0.0017), followed by a reduction across all groups thereafter (*p* = 0.003). CG displayed the lowest fibrinogen levels at 72 h (*p* = 0.006), which remained stable until the study endpoint (Table 3).

The evaluation of cervical morphology identified the papilla type as the most prevalent (33%, 10/30), followed by smooth (23%, 7/30), rosette (20%, 6/30), flap (17%, 5/30), and duckbill (7%, 2/30) (Table 4). It should provide a concise and precise description of the experimental results, the interpretation of the results, as well as the experimental conclusions that can be drawn.

## 4. Discussion

The pilot study proved efficient for hysteroscopic evaluation in non-estrous ewes (EPG). The caliber of the endoscope and protective sheath was effective for the examination. The technique using vaginoscopy was also efficient, without complications for the health of the ewes. Studies on goats using hysteroscopy techniques with a 4 mm rigid endoscope reported severe injuries, leading to adaptation with a 2 mm endoscope, which showed no complications [19]. For EPGp, the estrous mucus obstructed visualization; however, washing with solutions could be a simple option to improve visualization when accessing the uterus. However, solutions commonly used in mucus removal or lumen dilation, such as saline (NaCl 0.09%) or lactated Ringer’s solution, are spermicidal [20].

Even without intrauterine visualization, the AI technique via endoscopic vaginoscopy/hysteroscopy was promising, providing a more precise and higher-quality gynecological examination due to better animal restraint and potential cervical transposition, though not highly efficient in this study. In sheep, methods for better cervical visualization remain a challenge. A newly developed speculum specifically for ewes has demonstrated improved quality in gynecological examination [21]. Studies on cows have shown that endoscopic techniques improve overall examination quality, including cervix evaluation [9]. A major advantage of AI via endoscopic vaginoscopy is the absence of cervical traction, which has been reported as a factor that can cause reproductive tract injuries and raise concerns regarding animal welfare [22].

Plasma fibrinogen values remained within normal limits for sheep [23], demonstrating minimal invasiveness of the AI procedure overall. However, EVG exhibited higher fibrinogen values than the other two groups in the first two post-AI measurements, suggesting that vaginal insufflation with CO_2_ may have contributed to the increase. Nonetheless, this was not physiologically significant. According to Sabes et al. [24], fibrinogen serves as a sensitive indicator of inflammatory response in ruminants. The decrease in values at 72 h post-procedure in EVG was similar to observations in transcervical embryo collection, indicating that such procedures do not elevate fibrinogen levels beyond species-accepted limits [25]. Other studies show that cervical traction for AI affects cortisol levels in ewes, even under sedation, reinforcing the potential benefits of AI via endoscopic vaginoscopy as an alternative [26].

The procedure duration was longer in the pilot study’s endoscopic technique with estrus synchronization protocol, as mucus impaired visualization and consequently extended the procedure time. The other techniques had similar durations, suggesting that AI duration was not significantly influenced, as times were comparable to those reported in other AI studies using cervical traction and fixation [27]. Similarly to the cervical traction AI technique, the ability to observe the cervical ostium, whether through cervical traction or post-insufflation, is a favorable factor for insemination. This technique has been successfully used for intravaginal biopsy [28], although previous studies used lower CO_2_ pressure and flow rates than the present study. Silva et al. [29] used a pressure of 10 mmHg for vaginoscopy in a vaginal tumor resection procedure in a dog, demonstrating that the procedure is viable at this pressure. During the current study, no discomfort was observed in the animals, confirming that anesthesia was unnecessary for this AI technique.

The frequency of cervical ostium types in this experiment differed from the other literature reports, with papilla-type ostia being the most common and duckbill-type the least common. Franco et al. [30], using the same breed as this study, found that the duckbill-type ostium was more prevalent (46%). This discrepancy may be due to the absence of animal selection based on cervical type, which exhibits significant variability. Due to the lack of repeatability, as demonstrated by Moura et al. [31], cervical ostium transposition remains a challenge, making AI a highly complex procedure. The semen deposition site did not significantly influence pregnancy rates among the groups, but pregnancies were observed in ewes where cervical transposition was achieved. This result aligns with findings by Rekha et al. [32] and Duarte et al. [27], which demonstrated better efficiency for intrauterine AI, deep cervical AI, superficial cervical AI, and intravaginal AI, respectively. The semen deposition site did not significantly influence pregnancy rates among the groups, but pregnancies were observed in ewes where cervical transposition was achieved.

Overall, cervical transposition success and pregnancy rates may be influenced by several factors: inseminator skill, AI timing, and semen type. The practice and familiarity of the technician with the insemination method are directly related to pregnancy rates [32,33,34,35,36]. Another potential factor influencing the low pregnancy rates was the semen volume and processing method used [35]. The AI technique via endoscopic vaginoscopy showed good application, already allowing cervical passage without traction. This technique could yield better results if combined with cervical dilators, as demonstrated by Duarte et al. [27].

The efficiency of artificial insemination techniques in sheep is significantly influenced by factors such as climate, photoperiod, and the reproductive class of the females. The photoperiod, for instance, affects the reproductive activity of ewes, with shorter daylight periods stimulating estrus onset, while longer days may inhibit it. Furthermore, the reproductive class—distinguishing between nulliparous and multiparous females—impacts cervical anatomy, as nulliparous ewes typically present a narrower and more tortuous cervical canal, making transcervical access for procedures such as artificial insemination more difficult [37,38].

To overcome these limitations, several studies have investigated the use of pharmacological agents to induce cervical dilation and facilitate uterine access. However, even with such interventions, individual anatomical and physiological variability remains a challenge. In this context, vaginal endoscopy emerges as a promising technique, enabling direct visualization and more precise access to the cervical canal, regardless of individual variations. Despite its advantages, it is important to emphasize that the present study was conducted exclusively on medium-sized multiparous ewes, all submitted to fixed-time artificial insemination protocols, without evaluating the effects of photoperiod or reproductive class. Therefore, future studies are needed to assess the applicability and effectiveness of the technique under different conditions and animal categories [22,39].

Future investigations should explore strategies to improve cervical transposition rates, including the use of cervical dilators, pharmacological modulators of cervical relaxation, or new synchronization protocols tailored to endoscopic procedures. Comparative studies involving different breeds, cervical morphologies, and reproductive statuses may help identify key anatomical or physiological predictors of success. Additionally, further research is needed to assess the impact of these techniques on animal welfare, stress biomarkers, and long-term fertility outcomes. Refining instrumentation and standardizing training protocols for operators could also enhance reproducibility and field application. These efforts may contribute to consolidating endoscopic insemination as a viable tool in reproductive biotechnologies for small ruminants.

## 5. Conclusions

The use of transcervical endoscopic techniques in Santa Inês sheep proved to be feasible and safe for artificial insemination procedures. Videovaginoscopy enabled effective visualization of the cervix and intrauterine semen deposition in some animals without the need for cervical traction. Despite anatomical limitations, the pregnancy rates obtained were comparable between the evaluated methods, and the inflammatory markers remained within physiological limits, reinforcing the minimal invasiveness and clinical applicability of the technique. Although not yet highly efficient, endoscopic approaches already allow intrauterine access and represent a promising alternative for improving assisted reproduction in ruminants, particularly sheep.

## Figures and Tables

**Figure 1 life-15-00846-f001:**
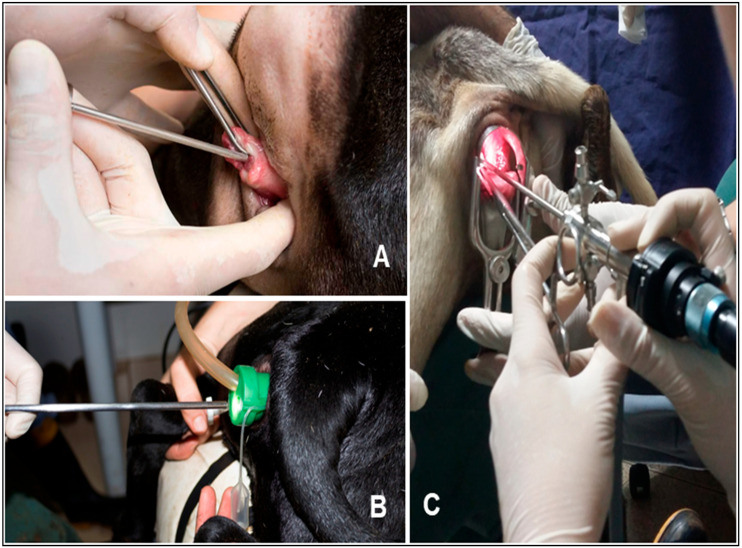
Transcervical artificial insemination techniques in Santa Inês sheep. Images: (**A**) transcervical AI technique with traction, Control Group—GC; (**B**) transcervical AI technique via endoscopic vaginoscopy, Vaginoscopy Group without cervical fixation with vaginal insufflation—GVe; (**C**) endoscopic AI technique with cervical traction, used in the Pilot Group without and with synchronization protocol—GPen and GPep.

**Table 1 life-15-00846-t001:** Comparison of reproductive and temporal parameters across groups.

Parameter	EPGp	EPG	CG	EVG	*p*-Value
Cervical passage rate (%)	10 (1/10)	100 5/5	–	–	<0.001
Intrauterine deposition (%)	10 (1/10)	10 (1/10)	20 (2/10)	20 (2/10)	0.0078
Pregnancy rate (%)	10 (1/10)	–	20 (2/10)	20 (2/10)	1.0
Insemination time (s)	249 ± 101	–	158 ± 50	129 ± 29	0.0017

**Table 2 life-15-00846-t002:** Number of cervical rings passed and semen deposition site.

Group	Cervical Rings (Mean ± SD)	Intrauterine (%)	Deep Cervical (%)	Superficial Cervical (%)
EPG	-	100 (5/5)	-	-
EPGp		10 (1/10)	-	90 (9/10)
CG	3.5 ± 3.3	20 (2/10)	80 (8/10)	–
EVG	1.6 ± 1.2	20 (2/10)	–	80 (8/10)

**Table 3 life-15-00846-t003:** Fibrinogen trends after insemination.

Time Interval	EVG Trend	CG Trend	*p*-Values
20 min to 24 h	Increase	–	0.0017
24 h to 72 h and beyond	Decrease	Decrease (lowest at 72 h)	0.003/0.006

**Table 4 life-15-00846-t004:** Distribution of cervical os types in the study population.

Cervical os Type	Frequency (n)	Prevalence (%)
Papilla	10	33 (10/30)
Smooth	7	23 (7/30)
Rosette	6	20 (6/10)
Flap	7	17 (7/10)
Duckbill	7	7 (2/30)

## Data Availability

The original contributions presented in the study are included in the article, further inquiries can be directed to the corresponding author.

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
