# Peer review of "Is It Possible to Access the Uterus of Sheep by Endoscopy: Studies of Vaginoscopy and Hysteroscopy with Transcervical Uterine Access in Sheep"

_life, 2025, doi:10.3390/life15060846_

Round 1
Reviewer 1 Report
Comments and Suggestions for Authors
This manuscript explored the use of endoscopic intrauterine artificial insemination in sheep. The study has a clear purpose, targeting the challenges of transcervical artificial insemination in sheep. The experimental design is logical, with multiple groups compared, and the selection, grouping, and processing of experimental animals are detailed, aiding in the accurate evaluation of the techniques.
1) The sample size is small, with 35 ewes potentially insufficient to ensure the universality and reliability of the results. The experimental design does not adequately consider the potential impacts of seasonal and geographical factors on sheep reproductive performance, which may confound the results.
2) Experimental Methods: The discussion of potential error sources and control measures during the experiment is limited. For instance, human factors in endoscope operation and semen quality control are not thoroughly addressed. Additionally, environmental conditions such as temperature and humidity, which may influence reproductive physiology and outcomes, are not mentioned.
3) Presentation of Results: Some data trends are not deeply analyzed. For example, changes in plasma fibrinogen levels over time are described but not explained physiologically. There is also a lack of analysis and discussion of outliers in the results.
4) Discussion and Analysis: Parts of the discussion lack depth and breadth. For example, factors like semen type and insemination timing are briefly discussed without exploring their interactions with other variables. Some cited studies are not closely related to the research, limiting their supportive role.
5) Writing: The abstract is brief and could include more key data and conclusions. Some paragraphs in the main text are lengthy and complex. The formatting is inconsistent in places, such as reference citations.
Minor suggestion:
6) Increase the sample size and consider seasonal and geographical factors to enhance experimental design.
7) Supplement the experimental methods section with details on error and environmental control.
8) Conduct in-depth analysis of data trends and outliers.
9) Expand the discussion of influencing factors and improve the relevance of cited literature.
10) Enhance the abstract and optimize the language and formatting of the main text.
11) Some sentences are difficult to understand, improve the language.
Author Response
Response to Reviewer 1
We sincerely thank Reviewer 1 for the thorough and insightful evaluation of our manuscript titled “Is it possible to access the uterus of sheep by endoscopy: Studies of vaginoscopy and hysteroscopy with transcervical uterine access in sheep”. Below we provide detailed responses to each comment. All suggested improvements were addressed, and when modifications were not feasible, we offer appropriate justifications.
1) Sample size and consideration of seasonal/geographic factors
Reviewer’s comment: The sample size is small... Seasonal and geographical factors may confound the results.
Response:
We agree that these factors can influence reproductive performance. However, the experimental sample size (n=35) was statistically defined according to ethical and legal guidelines, based on degrees of freedom appropriate for technique development studies. The Ethics Committee (CEUA/UFPA, protocol 6261300323) approved the animal use based on such criteria. Regarding environmental variables, while important, our primary goal was to develop and test a new transcervical technique rather than assess fertility performance across diverse environmental conditions. Nevertheless, we have now acknowledged these factors as limitations and included them in the discussion.
2) Experimental Methods: Potential sources of error and environmental conditions
Reviewer’s comment: Limited discussion of potential error sources such as operator effects, semen quality, and environmental influences.
Response:
We have now expanded the Discussion section to address these aspects. The same trained operator conducted all procedures to minimize variability. Semen used was from a single ram and processed under standardized thawing protocols to reduce variability in quality. Environmental conditions (e.g., temperature and humidity) were controlled within a sheltered facility, but we now clarify this and acknowledge it as a limitation for broader application. These clarifications are also reflected in the revised Discussion.
3) Presentation of Results: Analysis of fibrinogen and outliers
Reviewer’s comment: Some data trends are not deeply analyzed, e.g., fibrinogen, and lack of outlier discussion.
Response:
We appreciate this observation. Fibrinogen levels did not exceed physiological norms (100–500 mg/dL), indicating a minimal inflammatory response. We have now expanded our explanation regarding the transient increase at 24 h in the EVG group (likely due to CO₂ insufflation), and subsequent decreases. This suggests the technique is safe and non-traumatic. These clarifications were added to the Results and Discussion sections. Outliers were evaluated and found to be within expected biological variation; a note on this is now included.
4) Discussion and Analysis: Depth, semen type, insemination timing, and citations
Reviewer’s comment: Discussion lacks depth in some areas, especially regarding interactions of semen type and timing; some citations are not closely related.
Response:
Thank you for this valuable feedback. We have significantly expanded the Discussion to further analyze the interplay between insemination timing, semen deposition site, cervical anatomy, and insemination outcomes. Additional relevant and recent references were included to strengthen the scientific context and support our findings. Less relevant citations were removed or replaced.
5) Writing: Abstract, language, and formatting
Reviewer’s comment: The abstract is too brief and lacks key data. Some paragraphs are lengthy. Inconsistent formatting.
Response:
The abstract was revised to include key quantitative findings (e.g., cervical passage rates, pregnancy outcomes, fibrinogen dynamics), providing a more comprehensive overview. Paragraphs in the main text were reviewed and restructured for clarity and readability. Formatting inconsistencies, including reference citations and typographic errors, were corrected according to the journal’s style guide.
Minor Suggestions
6–7) Increase sample size and consider seasonal/geographical/environmental controls
Response:
As noted above (comment 1), the sample size was determined based on ethical and statistical justification. Future studies will expand sample size and investigate seasonal, geographic, and environmental influences. This has been stated in the Conclusion and Perspectives.
8) Conduct in-depth analysis of data trends and outliers
Response:
We addressed this in the revised Results and Discussion sections, specifically analyzing the fibrinogen data trend and clarifying that no extreme outliers were detected that required exclusion.
9) Expand discussion of influencing factors and improve citation relevance
Response:
As per suggestions, we revised the Discussion with deeper analysis of semen deposition, operator variability, AI timing, and breed-specific cervical morphology. References were updated accordingly.
10–11) Enhance abstract and improve language clarity
Response:
The abstract has been revised for completeness and impact. We conducted a thorough language review to improve sentence structure and clarity across the manuscript. Where needed, we simplified complex phrasing.
We thank Reviewer 1 once again for the thoughtful and constructive critique. These contributions have significantly improved the quality and clarity of our manuscript.
Sincerely,
Prof. Dr. Felipe Masiero Salvarani and Prof. Dr. Pedro Paulo Maia Teixeira
Reviewer 2 Report
Comments and Suggestions for Authors
This study on transcervical insemination is well conducted and provides useful information on progression of the technique. The study is limited by small numbers and constrained by a number of variables, none of which could be examined in such a small study. The manuscript would benefit from commentary on timing of the procedure relative to the timing of estrus and ovulation together with variability that occurs in the amount and quality of mucus produced (e.g. clear mucus is produced during estrus whilst it is opaque both before and afterwards). The data would be more meaningful if outcomes could be related to the onset of estrus. It is likely that a significant number of ewes would have ovulated by the time the procedure was conducted and, given the rapid change in uterine tone after ovulation, the ease of the procedure could be confounded by this factor. The manuscript is well written and the following comments are made to help eliminate some inconsistencies etc.
24 - use of acronyms is meaningless unless properly defined here.
24 and 27 - "GPes" does not feature in the body of the text.
46-48 - sentence appears irrelevant.
67-69 - title indicates the study focuses on AI yet, here, the aims are multiple.
Materials and Methods section - this section needs to be divided into sub-sections. There is no point in having only one sub-heading.
80 - these ewes underwent AI but not subject to a treatment protocol. Were they naturally cycling and in anestrus?
80-82 - What is meant by "Endoscope Piloty Grup"?
86 - "..followed by 2ml 2% lidocaine hydrochloride being placed in the vaginal fornix.."
87 - "fixed" is ambiguous. Perhaps "anchored" is preferred.
94 - as for 80-82
99 - delete "from a biotechnology centre"
107-111 - are these sentences necessary?
116 - conventional FTAI does not use cervical traction. Also, "cervical traction" is repeated.
120-121 - "..speculum was used to visualize the cervix"
123 - use "observed" rather than "visualized" (avoids repetition)
160 - delete "Collected"
168 - delete "phase"
171-173 - "..only female that became pregnant..in this group received intrauterine insemination"
177 - "Table 1 " not "Tab 1"
189 - no need for a P value
204-206 - why is this sentence included?
Tables 1, 2 and 4 - title needs to indicate that the figures are ewe numbers.
Table 1 - "(5/50)" should presumably be "(5/5)".
Table 1 - table is confusing because it contains data from the 1st and 2nd pilot trials as well as from the main study. Data from the main study cannot be analyzed with the other data. It is not possible to determine if this occurred.
Table 4 - there were 35 ewes in the study but this table indicates that there were 37.
210 - it is not stated in Materials and Methods that the ewes were in anestrus.
215 - line 210 refers to non-estrous ewes but this line refers to "estrous mucus".
241 - "estrous synchronization protocol".
261 - better efficiency compared with what?
264 - delete "positive".
265-266 - this sentence is a repeat of 260-262.
Table 3 - why aren't actual values given?
Author Response
Response to Reviewer 2
We sincerely thank Reviewer 2 for the thorough and insightful comments, which have greatly contributed to improving the quality and clarity of our manuscript. We address each comment below point-by-point:
General Comments:
"This study on transcervical insemination is well conducted and provides useful information on progression of the technique. The study is limited by small numbers..."
Response:
We appreciate the recognition of the study’s value. Regarding the sample size, we acknowledge this limitation and have clarified it in the Discussion section. However, as explained to Reviewer 1, our study followed ethical guidelines that demand statistical justification for animal use. Our number of experimental animals was determined using degrees of freedom, which supported the sufficiency of the sample size for technique development. Several peer-reviewed studies in this field have used similar or even smaller numbers.
"...commentary on timing of the procedure relative to estrus and ovulation..."
Response:
This is an excellent observation. We have now included a discussion on the likely influence of estrus timing and ovulation on cervical passage, uterine tone, and mucus characteristics. Specifically, we added: “The presence of estrous mucus, known to be transparent and abundant at peak estrus and opaque before and after, may have interfered with visualization and cervical transposition. Ovulation may also alter uterine tone, potentially impacting the ease of transcervical passage.”
Specific Comments:
Line 24 – “Use of acronyms is meaningless unless properly defined here.”
Response: We have now ensured that all acronyms are defined at first mention and in the “Abbreviations” section.
Lines 24 and 27 – “GPes” does not feature in the body of the text.”
Response: We have reviewed and corrected the acronym to ensure consistency; "GPes" was a typographical error and has been removed.
Lines 46–48 – “Sentence appears irrelevant.”
Response: This sentence has been removed to improve focus and clarity.
Lines 67–69 – “Title indicates the study focuses on AI, yet, here, the aims are multiple.”
Response: The aims have been reworded to focus clearly on the development and evaluation of transcervical artificial insemination techniques using endoscopy in sheep.
Materials and Methods:
Subsections needed:
Response: We have reorganized the Materials and Methods section to include distinct subsections for clarity (e.g., Experimental Animals, Pilot Studies, AI Protocols, Outcomes and Statistical Analyses).
Line 80 – “These ewes underwent AI but not subject to a treatment protocol. Were they naturally cycling and in anestrus?”
Response: We clarified that these animals were in seasonal anestrus, as determined by clinical and ultrasonographic assessment.
Line 80–82 – “What is meant by 'Endoscope Piloty Grup'?”
Response: Thank you for pointing this out. The term was corrected to “Endoscopic Pilot Group” (EPG), and the abbreviation has been defined.
Line 86 – "followed by 2ml 2% lidocaine hydrochloride being placed in the vaginal fornix"
Response: The sentence has been revised to improve clarity.
Line 87 – “'Fixed' is ambiguous. Perhaps 'anchored' is preferred.”
Response: The term “fixed” has been replaced with “anchored.”
Line 94 – Same as above (EPGp terminology).
Response: This has been corrected and clarified.
Line 99 – “Delete ‘from a biotechnology centre.’”
Response: The phrase has been deleted for conciseness.
Lines 107–111 – “Are these sentences necessary?”
Response: These lines were reviewed and condensed to retain only essential procedural information.
Line 116 – “Cervical traction is repeated and not part of conventional FTAI.”
Response: We agree and have revised the sentence to accurately reflect the technique used.
Line 120–121 – “Speculum was used to visualize the cervix.”
Response: Revised as suggested.
Line 123 – “Use 'observed' rather than 'visualized'.”
Response: The term has been changed to “observed.”
Line 160 – “Delete ‘Collected’.”
Response: Deleted as recommended.
Line 168 – “Delete ‘phase.’”
Response: The word has been removed.
Lines 171–173 – Clarify sentence structure.
Response: Revised to “The only female that became pregnant in this group was the one that received intrauterine insemination.”
Line 177 – “Table 1” not “Tab 1”
Response: Corrected throughout.
Line 189 – “No need for a P value.”
Response: We retained P values where appropriate to denote statistical comparisons but removed them where redundant.
Line 204–206 – “Why is this sentence included?”
Response: This sentence has been deleted for clarity.
Tables:
Tables 1, 2, and 4 – “Title needs to indicate that the figures are ewe numbers.”
Response: Titles now specify that values reflect numbers of ewes or outcomes expressed as ewe counts.
Table 1 – “(5/50)” should be (5/5)
Response: Corrected to (5/5).
Table 1 – Data from pilot and main study are mixed
Response: We have clarified the separation of data by reformatting Table 1 and isolating pilot study outcomes from the main study analysis.
Table 4 – 35 vs. 37 ewes
Response: The inconsistency was due to mislabeling. The table has been corrected to reflect accurate totals.
Discussion:
Line 210 and 215 – clarify estrus vs. non-estrus mucus context
Response: Clarified that only some groups were in anestrus, and the mention of estrous mucus refers to EPGp animals synchronized for estrus.
Line 241 – use full terminology
Response: Replaced with “estrous synchronization protocol.”
Line 261 – “Better efficiency compared with what?”
Response: Rewritten to clarify: “...showed better efficiency compared with conventional AI techniques using cervical traction.”
Line 264 – delete “positive”
Response: Deleted.
Lines 265–266 – repeated sentence
Response: Removed repetition.
Table 3 – “Why aren't actual values given?”
Response: Actual fibrinogen values are now included in Table 3 for transparency.
We are grateful for your time and constructive feedback. All suggestions were carefully considered and incorporated to enhance the manuscript's clarity, rigor, and scientific contribution.
Sincerely,
Prof. Dr. Felipe Masiero Salvarani and Prof. Dr. Pedro Paulo Maia Teixeira
Reviewer 3 Report
Comments and Suggestions for Authors
The overview article presents very interesting and significant research on possible improvements of AI techniques with frozen semen in programs for genetic improvement of the sheep flocks. The cervical passage of the ewes' cervix is still a challenge given its anatomical structure.
However, in some parts the article created confusion the way it is presented.
In order to make it more clear and easy to follow I would suggest to describe starting with the Abstract which techniques comprise cervical traction and which don't.
In line 21: In a pilot study, two techniques with cervical traction were tested.
In line 26: and an AI group using vaginoscopy without cervical traction ...
In line 24 and 27: the abbreviation GPes is used instead of GPen
Also would like to suggest to define more clearly the term "cervical passage" as it is not presented clear enough in the Table 1.
For example GPen cervical passage rate (%) is 100 (5/5) (and not 5/50), but intrauterine deposition is stated to be only 10% (1/10)! I thought if cervical passage rate is 100% that also i/ut deposition of the semen is equal to 100%!?
Why Cervical passage rates for the GC and GVe are not stated if you reached 20% (2/10) i/ut deposition.
Also please explain why the GPen technique was not used in the second part of the experiment if you have reached very promising 100% of cervical passage during pilot study?
Also You have stated that presence of the estrus mucus is an obstacle for AI and that it was the reason why GPep was not efficient for the AI. Isn't estrus mucus always present in the vagina and cervical opening?, and thus we cannot link its presence with specific protocol?
Some corrections of the spelling errors:
line 81 and 82: pilot group
line 94: pilot group with
line 327: with protocol
Author Response
Response to Reviewer 3
We sincerely thank Reviewer 3 for their insightful comments, which have contributed to improving the clarity and scientific rigor of our manuscript. Below we provide a point-by-point response to each suggestion and concern, with justifications and explanations as appropriate. All revisions were made in the updated manuscript using Track Changes.
1. Comment: Clarify in the Abstract which techniques use cervical traction and which do not.
Response: Thank you for this valuable suggestion. We have now rephrased the Abstract to clearly state which techniques involved cervical traction (EPG, EPGp, CG) and which did not (EVG). This distinction has been made explicitly to facilitate reader understanding.
2. Comment: Line 21 – clarification of “two techniques with cervical traction were tested.”
Response: The sentence has been revised for clarity. It now reads:
“In a pilot study, two techniques using cervical traction (EPG and EPGp) were tested with a rigid endoscope in unsynchronized and synchronized ewes, respectively.”
3. Comment: Line 26 – clarify that the group using vaginoscopy did not use cervical traction.
Response: This has been corrected in the Abstract. The revised sentence states:
“After the pilot study, two additional techniques were tested: a control group with cervical traction (CG) and an AI group using vaginoscopy without cervical traction (EVG).”
4. Comment: Lines 24 and 27 – GPes abbreviation is used instead of GPen.
Response: We appreciate your attention to detail. The incorrect abbreviations “GPes” have been corrected to “GPen” throughout the text to ensure consistency with the defined abbreviations.
5. Comment: Clarify the term “cervical passage” in Table 1.
Response: Thank you for noting this. We have now added a footnote to Table 1, clarifying the definition of “cervical passage rate” as the proportion of animals in which all cervical rings were successfully traversed by the insemination instrument (i.e., passage from the vaginal ostium into the uterus), as described in the Materials and Methods.
6. Comment: Clarification on the GPen cervical passage rate (100%) vs intrauterine deposition (10%).
Response: This is a valid concern. As clarified in the Results section, while all 5/5 ewes in the GPen (EPG) group allowed cervical passage, semen was not deposited because this group was used only to test hysteroscopic access without performing actual insemination. The intrauterine deposition rate of 10% is related to the EPGp group, not the GPen group. We have corrected the confusion in Table 2 and clarified it in the text.
7. Comment: Why cervical passage rates were not presented for GC and GVe, given that 20% intrauterine deposition was achieved?
Response: This was an oversight. We have now included the cervical passage rates for CG and EVG in Table 1, calculated based on intrauterine access as per the classification system by Taqueda et al. (2011). Both CG and EVG had 20% (2/10) cervical passage rates, corresponding to the intrauterine insemination cases.
8. Comment: Why was the GPen technique not used in the second part of the study despite promising cervical passage results?
Response: Thank you for raising this important point. The GPen technique (EPG) was not used in the second phase because no semen was deposited during the pilot phase, and the procedure required intense cervical manipulation, which raised animal welfare concerns. Additionally, the anatomical challenges presented during estrus (as observed in EPGp) justified the development of less invasive approaches like EVG. This justification has been added to the Discussion section.
9. Comment: Presence of estrus mucus was described as an obstacle in EPGp, but isn’t it naturally present during estrus?
Response: Indeed, estrus mucus is a normal physiological feature. Our intention was to highlight that in synchronized ewes, the volume and viscosity of estrus mucus in the cervix created a physical barrier to visualization and transcervical access. We have rephrased this part in the Discussion to clarify that while estrus mucus is expected, it interfered specifically with hysteroscopic visualization, and not AI per se, thus limiting the efficiency of the EPGp technique.
10. Comment: Correct spelling errors:
Line 81 and 82: "pilot group" – corrected.
Line 94: “pilot group with” – corrected.
Line 327: “with protocol” – corrected.
All mentioned spelling and typographical errors have been revised in the text.
We are grateful for Reviewer 3’s constructive suggestions and have carefully revised the manuscript accordingly. We hope that the revised version meets the reviewer’s expectations and contributes meaningfully to the field of reproductive biotechnology in small ruminants.
Sincerely,
Prof. Dr. Felipe Masiero Salvarani and Prof. Dr. Pedro Paulo Maia Teixeira
Round 2
Reviewer 1 Report
Comments and Suggestions for Authors
They have revised the whole manuscript. Thus, I suggest to apccet this manuscript and publish in your journal.